# Toward Exascale Climate Modelling: A Python DSL Approach to ICON's (Icosahedral Non-hydrostatic) Dynamical Core (icon-exclaim v0.2.0)

Anurag Dipankar[1], Mauro Bianco[3], Mona Bukenberger[2], Till Ehrengruber[3], Nicoletta Farabullini [1], Oliver Fuhrer[4], Abishek Gopal[5], Daniel Hupp[4], Andreas Jocksch[3], Samuel Kellerhals[1], Clarissa A Kroll[2], Xavier Lapillonne[4], Matthieu Leclair[1], Magdalena Luz[1], Christoph Müller[4], Chia Rui Ong[1], Carlos Osuna[4], Praveen Pothapakula[2], Andreas Prein[2], Matthias Röthlin[4], William Sawyer[3], Christoph Schär[2], Sebastian Schemm[7], Giacomo Serafini[4], Hannes Vogt[3], Ben Weber[4], Robert C. Jnglin Wills[2], Nicolas Gruber[6], Thomas C. Schulthess[3]

[1]Center for Climate System Modelling C2SM, ETH Zürich, Switzerland
[2]Institute for Atmospheric and Climate Sciences IAC, ETH Zürich, Switzerland
[3]Swiss National Supercomputing Centre CSCS, ETH Zürich, Switzerland
[4]Federal Office of Meteorology and Climatology MeteoSwiss, Switzerland
[5]NSF National Center for Atmospheric Research, Boulder, Colorado
[6]Institute of Biogeochemistry and Pollutant Dynamics, ETH Zürich, Switzerland
[7]Department of Applied Mathematics and Theoretical Physics, Cambridge University, UK

*Correspondence to:* Anurag Dipankar (anurag.dipankar@c2sm.ethz.ch)

## Abstract.

A refactored atmospheric dynamical core of the ICON model implemented in GT4Py, a Python-based domain-specific language designed for performance portability across heterogeneous CPU-GPU architectures, is presented. Integrated within the existing Fortran infrastructure, the new GT4Py dynamical core is shown to exceed ICON OpenACC performance. A multi-tiered testing strategy has been implemented to ensure numerical correctness and scientific reliability of the model code. Validation has been performed through global aquaplanet and prescribed sea-surface temperature simulations to demonstrate model's capability to simulate mesoscale and its interaction with the larger-scale at km-scale grid spacing. This work establishes a foundation for architecture-agnostic ICON global climate and weather model, and highlights poor strong scaling as a potential bottleneck in scaling toward exascale performance.

## 1 Introduction

Simulating Earth's atmosphere at a horizontal grid spacing of a few kilometres, commonly referred to as storm-resolving simulations, marks the first step towards a new regime of climate and weather modelling. At such grid spacing, assuming that the vertical grid spacing is similarly adjusted, one expects the interactions between the mesoscale and the larger scales to be represented more accurately than at a grid spacing of tens of kilometres. This expectation is well justified, knowing that at km-scale such interactions are not influenced by empirical parameterisation but are instead governed dynamically.

The Nonhydrostatic Icosahedral Atmospheric Model (NICAM; Satoh et al., 2014) modelling group started working towards global storm resolving simulations on the Japanese K-computer (Satoh et al., 2017), which then paved the way for further developments. The larger community joined the efforts towards global storm resolving simulations using a horizontal grid spacing of less than or equal to 5 km in the second phase of DYnamics of the Atmospheric general circulation Modelled On Non-hydrostatic Domains (DYAMOND Winter; Duras et al., 2021). 9 of the 12 models contributing to DYAMOND Winter use a grid spacing of 5 km or less. Only 2, ARPEGE-nh (Action de Recherche Petite Echelle Grande Echelle Non-Hydrostatic; Bubnovà et al., 1995) and ICON (ICOsahedral Non-hydrostatic; Zängl et al., 2015) of these 9 models employ a grid spacing of 2.5 km or less. The progressively decreasing number of models with decreasing grid spacing in DYAMOND Winter underscores the challenge towards km-scale modelling-that it is a complex scientific-technical problem. The scientific challenge lies in adjusting models numeric and the remaining sub-grid scale parameterisation to a very new regime that is free of an important and highly tuneable convection parameterisation. The technical challenge lies in making these models run reliably and efficiently at scale on the modern computing systems.

We focus here on performance and reliability of these models. Achieving performance good enough to be able to simulate a few decades in a reasonable time is probably the most commonly discussed challenge. One simulation year per computational day is often used as a benchmark in this regard (Schulthess et al., 2019). Reliability, defined as the ability to run a model stably at scale on a supercomputer, is a known but less frequently published challenge, often confined to discussions in specialised workshops and conferences. We begin by addressing performance.

Stevens et al. (2019) reported that models participating in the first phase of DYAMOND typically produced six simulation days per computational day (SDPD) on grid spacing of roughly 2.5 km, amounting to 0.75 SDPD on a grid spacing of 1.25 km, which is about a factor 500 short of the goal of 365 SDPD. In the last six years, the community has invested substantial resources in performance optimizing their codes. The recent numbers are very encouraging. Klocke et al. (2025), see their Table 1 for a detailed assessment, report 26 SDPD for the Simple Cloud-Resolving E3SM Atmosphere Model (SCREAM; Donahue et al., 2024), 17 SDPD NICAM, and 145.7 SDPD for ICON. Obviously, these numbers are not comparable since the models were run on different machines using different number of compute tasks but an increasing trend is clearly seen.

While these performance numbers are encouraging, they remain insufficient for any practical tuning and multi-decadal production runs at 1–2 km grid spacing. GPUs offer significant acceleration but suffer from poor strong scaling (Giorgetta et al., 2022; Adamidis et al., 2025). CPUs, by contrast, scale well but incur high energy costs, which is undesirable (Adamidis et al., 2025). Given the rapid evolution of computing architectures, it is unwise to tie models to a specific platform (Schulthess, 2015). A logical solution is to adopt the principles of Domain Specific Languages (DSLs)—specifically, separation of concerns. This allows user code to remain unchanged while DSL abstractions enable backend flexibility across architectures.

This view is shared by several modelling groups: developers of the Portable Model for Multi-Scale Atmospheric Prediction (PMAP; Ubbiali et al., 2025) and PACE (Dahm et al., 2024) use the Python-based DSL GT4Py (Paredes et al., 2023). SCREAM employs the C++ library Kokkos (Trott et al., 2022), and the UK Met Office uses the Fortran-based DSL PSyclone (Pysclone, 2025) for its next-generation modelling system.

However, performance metrics alone do not reflect the reliability of computing platforms required for large-scale simulations using thousands of GPUs. Experience and discussions in workshops

(e.g., iCAS24, Hart 2024) suggest that global km-scale modelling is cutting-edge not only in application but also in hardware and interfacing libraries. Node failures, memory overflows, filesystem crashes, and random errors are significantly more frequent when simulating at scale. These issues are more apparent for the legacy codes because Fortran compiler support on the newer architectures is being increasingly deprioritized by the vendors. Addressing these is difficult, given the continuous evolution of hardware and software. Nevertheless, application developers can improve reliability by testing code at various granularities to ensure robustness of both the application and the underlying platform.

Within EXtreme scale Computing and data platform for cLoud-resolving weAther and clImate Modeling (EXCLAIM), we are addressing the aforementioned challenges. The project seeks to develop a modularised code based on ICON that is performant, architecture agnostic, and at the same time reliable. Laid out as a three-phase development project (see Section 3), the present manuscript reports on Phase I: embedding GT4Py-based atmospheric dynamical core kernels into the existing Fortran framework. This integration achieves competitive performance relative to the original implementation, marking a critical step toward a fully modular and scalable system.

The manuscript is accordingly organised as follows. The details of the model and the DSL is given in section 2. The software development strategy of EXCLAIM and the placement of the current version in the roadmap is discussed in section 3 followed by details on code refactoring and testing strategies in section 4. Computational performance of the current version and future possibilities are presented in section 5. Simulation results from the scientific experiments are discussed in section 6. The manuscript ends with a conclusion in section 7.

## 2   The model and the new user code

The model is based on the global weather and climate modelling system ICON which is written primarily in Fortran. ICON is used for a large set of applications ranging from large-scale climate dynamics (Hohenegger et al., 2023; Giorgetta et al., 2018) to numerical weather prediction (Zängl et al, 2015; Prill et al., 2023) to large-eddy simulation (Dipankar et al., 2015; Heinze et al., 2017). All these applications share the same dynamical core and tracer advection routines but differ on the suite of physical parameterisations and their coupling technique to the dynamical core. While ICON has traditionally been used on homogeneous computing platforms using MPI and OpenMP parallelization, recent developments (Giorgetta et al., 2022 and Lapillonne et al, 2025) have made it work on heterogeneous CPU-GPU platforms using OpenACC directives. These developments are now used for global storm resolving simulations in the projects like nextGEMS (Segura et al., 2025), Destination Earth (Bauer et al., 2021), EXCLAIM, and for operational limited-area weather forecasts at the Swiss National Meteorological Service (Lapillonne et al., 2025).

The ICON dynamical core (see Zängl et al., 2015 for details), as any other dynamical core, is a complex and very large piece of code coupled to the physics and model infrastructure. In a typical ICON atmosphere-only simulation, the dynamical core is the most computationally expensive component, accounting for roughly 40% of the total cost. This part of the code also does not change much in time, making it the perfect first candidate to refactor for heterogeneous computing.

The refactored code is written in GT4Py, which is a Python-based embedded domain specific language for climate and weather modelling. GT4Py is developed at ETH Zürich together with the users. The users, other than EXCLAIM, include the developers of PMAP (Ubbiali et al., 2025) and PACE (Dahm et al., 2023).

129

130

131

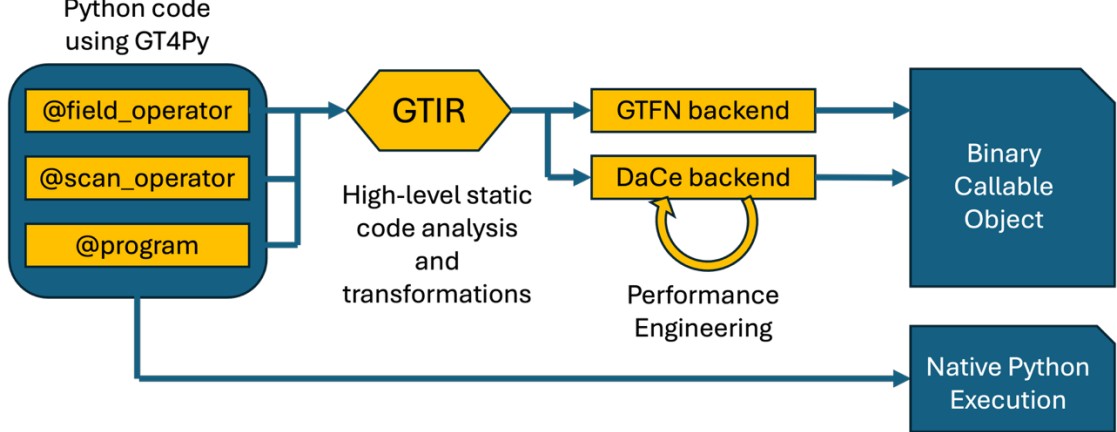

132

Figure 1 Schematic illustrating the various steps within GT4Py from the user front end to the executable. Here, GTIR is the GridTools Intermediate Representation; GTFN backend is the GridTools Fortran backend; DaCe backend is the Data-Centric programming paradigm backend.

GT4Py is comprised of (see Fig. 1) a user-facing interface, in which the computational patterns, like stencils, used in Climate and Weather applications can be easily composed. The main computations are captured by three concepts: `field_operator` to express operations on fields, `scan_operator` to express dependencies in the vertical direction, and `program` to compose the two. The high-level description of the computation is then taken automatically by the GT4Py parser where the code is translated into an intermediate representation (GTIR) and transformed with domain-specific high-level transformations and static code analysis to narrow down the code needed for the specific simulation at hand. The backend then takes the GTIR formulation of the computations to perform architecture dependent optimizations. GT4Py is designed for *portability of performance* and can generate code for NVIDIA and AMD GPUs, x86 and ARM CPUs. The user can select different backends, the native C++ GridTools GTFN backend (Afanasyev et al., 2021) or DaCe (Data-Centric programming paradigm, Ben-Nun et al., 2019). DaCe offer an open-box solution for optimization for finer tuning. It allows for performance engineers to tailor the optimization to the specific characteristics and semantics of the application, and possibly of the input configuration of the simulation.

GT4Py is designed to overcome the limitations of typical domain specific languages (DSLs), which usually involve offline compilation of computation kernels to be linked in program executables. DSLs like these usually simplify the writing of the kernels but make the integration into the main applications rather complex. GT4Py, instead, is embedded in Python and allows the users to write and execute the code directly in Python, either natively in Python or generating efficient codes using just-in-time compilation (JIT) or ahead-of-time compilation (AOT). The current implementation uses AOT.

## 3  Development roadmap

Due to the monolithic design of the Fortran-based ICON model, refactoring is both challenging and time-intensive. To ensure that scientific production and model development proceed in

parallel, we have adopted a development roadmap that is closely aligned with scientific use cases.
The core use cases include global aquaplanet simulations, global simulations with prescribed sea-
surface temperatures, and fully coupled global atmosphere-ocean simulations. Development is
structured in three phases each with one key deliverable tied to a core scientific use case, as
illustrated in Fig. 2.

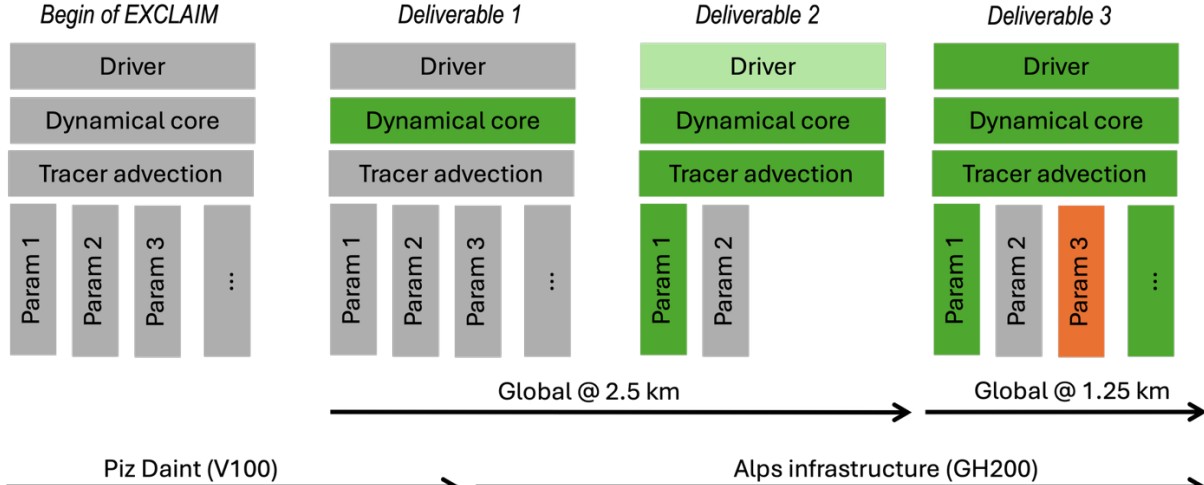


Figure 2. EXCLAIM's software development roadmap combined with timelines for the core scientific use cases
and the computational hardware. The colors indicate Programming languages: grey is Fortran+OpenACC,
green is GT4Py/Python, and orange is Kokkos.
Starting with the code version refactored using OpenACC, the first deliverable is the refactored
GT4Py dynamical core encapsulated within the original Fortran+OpenACC "Fortran+" ICON. This
initial deliverable is critical, as it establishes the foundation for continuous integration (CI) and
continuous deployment (CD) of the developed model. It also facilitates the transition from the
legacy Swiss National Supercomputing system, Piz Daint, to the new ALPS research infrastructure,
which utilises NVIDIA GH200. The goal of this first deliverable is to demonstrate the feasibility of
conducting a few years of global km-scale simulations on the ALPS infrastructure, with
performance comparable to the reference Fortran+ implementation.
The second deliverable underscores the transition towards a Python-based driver with
components in GT4Py. Adopting a Python driver will enable numerous features that would be
challenging in the traditional Fortran-based application. Firstly, the portability of the code would
be improved; secondly, the memory layout of the numerical fields could be adapted to the
architecture, thanks to the separation of concerns that GT4Py offers in decoupling data access
syntax from the actual data organization. Finally, the access to the vast Python ecosystem of
packages could greatly impact the usability of complex workflows and applications, beyond the
current capabilities. The driver is currently being developed (hence the light green color in Fig. 2)
and components such as tracer advection and microphysics are already implemented in GT4Py.
Since the driver is written from scratch, it is not expected to be fully featured to perform realistic
use cases. Instead, the second deliverable will be tested in idealised configuration(s).
The third and final deliverable is intended to enable realistic simulations thereby superseding the
first deliverable. The target horizontal grid spacing for use cases in this stage is 1.25 km globally.
The driver is designed to be modular, allowing integration of physical parameterisation schemes
written in languages other than GT4Py, for example, using Kokkos. The remainder of this
manuscript focusses on the first deliverable.

# 4   Code refactoring and testing

## 4.1   Code refactoring

Our starting point was the code base using Fortran+ as described in Giorgetta et al. (2022), which has been since further optimised (Lapillonne et al, 2025). The entire Fortran+ dynamical core, including numerical diffusion, is re-written except for the part involving Halo exchange. The original implementation is retained for the Halo exchange in the current version. Furthermore, the blocking length that is typically used in atmospheric codes for cache efficiency is not used in the refactored code.

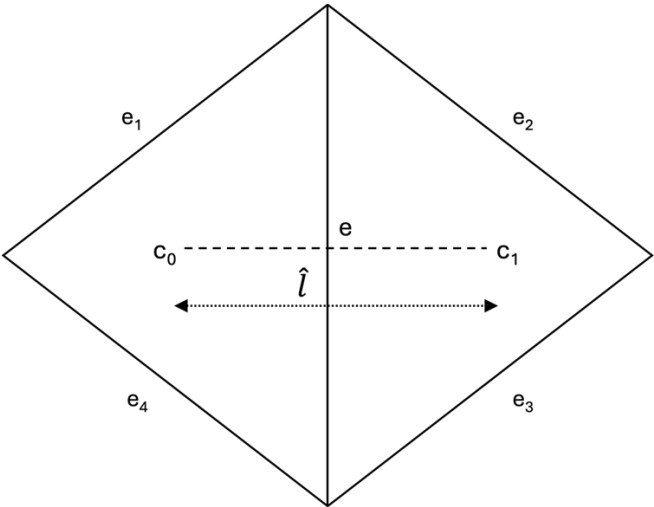

Figure 3. Schematic to illustrate gradient operation on a triangle edge as used in ICON. $e$ is the centre of the edge where operation is performed. The neighbouring edges are indicated from $e_1$ to $e_4$. Triangle cells about the edge $e$ are indicated by $c_0$ and $c_1$ with a separation of $\hat{l}$ between them.

We first demonstrate through a simple example of a computational stencil how the high-level description of computations in GT4Py simplifies the user code. Let's assume that a generic variable, $\psi$, is located at the centre of the edge (see Fig. 3) and we wish to compute its horizontal gradient in the direction normal to the edge using a centred difference. The mathematical expression reads as (Equation 1)

$$\frac{\Delta\psi}{\Delta c_0 c_1} = \frac{\psi_{c_1} - \psi_{c_0}}{\hat{l}} = \frac{\psi(c_1(e)) - \psi(c_0(e))}{\hat{l}} \tag{1}$$

where $c_0$ and $c_1$ are the cell centres of the given edge $e$, and $\hat{l}$ is the length between them . In Fortran, the above computation is written as shown in Listing 1.

```
1.  !$OMP PARALLEL
2.  !$OMP DO PRIVATE(jb, i_startidx, i_endidx, je, jk)
3.  DO jb = i_startblk, i_endblk
4.    CALL get_indices_e(ptr_patch, ...)
5.    !$ACC PARALLEL ....
6.  #ifdef __LOOP_EXCHANGE
7.    DO je = i_startidx, i_endidx
8.      DO jk = slev, elev
9.  #else
10.   DO jk = slev, elev
11.     DO je = i_startidx, i_endidx
12. #endif
13.       grad_norm_psi_e(je,jk,jb) =  &
              ( psi_c(iidx(je,jb,2),jk,iblk(je,jb,2)) -
                  psi_c(iidx(je,jb,1),jk,iblk(je,jb,1)) )
              / ptr_patch%edges%lhat(je,jb)
14.     ENDDO
15.   END DO
16. !$ACC END PARALLEL
17. END DO
18. !$OMP END DO NOWAIT
19. !$OMP END PARALLEL
```

Listing 1 Fortran+ code to compute equation 1 illustrating increasing complexity of the user code due to different pragmas for different parallelization methods

Clearly, a significant part of the code here is used to describe parallelization in OpenMP and OpenACC and for performance optimisation using loop exchange, which makes the code convoluted. Translation of the same code in GT4Py is shown in Listing 2. Here, the parallelization and performance details are not visible in the user code and the high-level description then allows one to write a code that is significantly simplified and easy to understand.

```
@field_operator
def _grad_norm(
    psi: Field[[CellDim, KDim], float],
    lhat: Field[[EdgeDim], float],
) -> Field[[EdgeDim, KDim], float]:
    return (psi(E2C(1)) - psi(E2C(0)))/lhat
```

Listing 2. Translation in GT4Py of the Fortran+ code in Listing 1 using a field operator for gradient compuation. CellDim and EdgeDim are the horizontal dimensions of triangular cells and edges. KDim is the vertical dimension. E2C points to the two cell centres about the edge.

The unstructured ICON grid often requires computations involving shifts between cells, edges, and vertices with subsequent summation over newly defined offsets. Example from the code in Listing 3

```
psi (iqidx(je,jb,1),jk,iqblk(je,jb,1)) +

psi (iqidx(je,jb,2),jk,iqblk(je,jb,2)) +

psi (iqidx(je,jb,3),jk,iqblk(je,jb,3)) +

psi (iqidx(je,jb,4),jk,iqblk(je,jb,4))
```

Listing 3 an example illustrating typical neighbour access in icon. here, an edge variable `psi` is summed over the four boundary edges in Figure 3.

represents the offset of the $\psi$ field from edges -> cells -> edges over the four edges ($e_1$ to $e_4$) in Fig. 3. On the other hand, GT4Py does not require indices specification or loops over dimensions bounds. This allows for a cleaner and intuitive solution: `neighbor_sum (psi (E2C2E))`

Not having the loops in GT4Py is one of its most fundamental features, since it allows for decoupling the mathematical expression from the data layout and the scheduling of the instructions on the architecture. This comes at the cost of some limitations in the expressiveness of the GT4Py as the user interface. This is why GT4Py is a domain-specific solution for weather and climate computations and not a generic framework for arbitrary arguments. With respect to other approaches, GT4Py builds upon several years of experience with different implementations (STELLA (Gysi et al., 2015) and GridTools (Afanasyev et al., 2021)), addressing not only expressing abstractly domain specific concepts but also stressing portability of performance and the integration into larger application frameworks.

Finally, even with the presence of some limitations, this work brought major improvements in comparison to the Fortran+ version: the code itself is more readable and allows for an easy local documentation through docstrings, and it is slightly superior in terms of performance.

## 4.2   Insertion of stencils with Liskov preprocessor

The ICON dynamical core consists of approximately 60 stencils which have horizontal dependencies through neighbouring cells, edges and vertices. These stencils were translated into GT4Py and then unit tested individually.  The requirement for their subsequent integration into ICON was that each could be verified with respect to the existing Fortran+ code. This approach requires extensive boilerplate, which would have resulted in unclean code. From the outset, it was clear that a preprocessing stage would be necessary to simplify this insertion task.

ICON Liskov[1] is a directive-based preprocessor which parses comments and substitutes them with code, facilitating the integration of the GT4Py generated code into the ICON model. A simple example illustrating the insertion of a stencil used in the horizontal numerical diffusion is depicted in Listing 4.

---

[1] Named after Barbara Liskov, an American computer scientist and Turing Award laureate who has made pioneering contributions to programming languages.

```
!$DSL START STENCIL( name=mo_nh_diffusion_stencil_10;
!$DSL               w=p_nh_prog%w(:,:,1);
!$DSL               diff_multfac_n2w=diff_multfac_n2w(:);
!$DSL               cell_area=p_patch%cells%area(:,1);
!$DSL               z_nabla2_c=z_nabla2_c(:,:,1);
!$DSL               vertical_lower=2;
!$DSL               vertical_upper=nrdmax(jg);
!$DSL               horizontal_lower=i_startidx;
!$DSL               horizontal_upper=i_endidx )
DO jk = 2, nrdmax(jg)
 DO jc = i_startidx, i_endidx
      p_nh_prog%w(jc,jk,jb) = p_nh_prog%w(jc,jk,jb) + &
          diff_multfac_n2w(jk)  * patch%cells%area(jc,jb)* &
          z_nabla2_c(jc,jk,jb)
  ENDDO
ENDDO
```

Listing 4. Example showing the use of Liskov in the Fortran code to generate corresponding GT4Py code.

Liskov can generate code in two modes: one for straightforward *substitution* and one for *verification*. The former simply inserts the appropriate GT4Py stencil, compiled ahead-of-time, instead of the intervening Fortran code. The latter instead executes both the stencil and the Fortran code and compares the results, giving an error message if they do not meet a given tolerance. These modes are illustrated in Fig. 4. While the verification mode was used extensively during development, the substitution mode has been used for the scientific testing and benchmarks presented in the following sections.

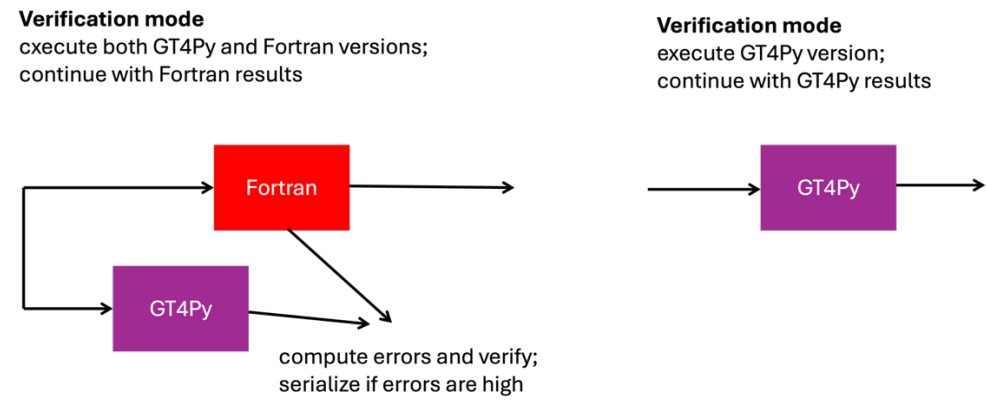

Figure 4: Liskov verification and substitution modes.

The Liskov preprocessor is only a temporary tool: for the longer term we are coalescing stencils into larger kernels to exploit data reuse.  These fused stencils have been incorporated into a full GT4Py dynamical core, which can be called independently from a Python (see deliverable 2 or 3) or the current Fortran driver. The work of creating a Fortran-callable interface for this dynamical core version is now complete, and an effort to optimize the ICON model version which calls this version is now ongoing.  This will constitute the final dynamical core product, and we will report on its features and performance in subsequent publications.

## 4.3  Testing

The ICON modelling system has its own testing infrastructure that runs a set of experiments on several machines. All the experiments go through a series of tests that have been commonly used

in the community. These include, for example, an *MPI test* to check the correctness of MPI parallelization, a *nproma test* to check the correctness of the implementation of horizontal loop blocking, a *restart test* to check the correctness of restart functionality, etc.

When porting a code for accelerated computing, testing is tricky as the results are different due to rounding. Lapillonne et al. (2025) have described *probtest* wherein the outputs from a GPU binary of a full integration are compared against that of a CPU binary with some tolerance. This test is part of the ICON testing infrastructure, and we have adopted it in our testing infrastructure as well. In addition, we include tests at fine granularity to ensure improved reliability of the application. Essentially, the refactored code is submitted to testing at three broad levels:

1.  Level 1 one-to-one testing between individual Fortran stencils and their GT4Py counterparts at a coarser grid spacing.
2.  Level 2 perturbation growth test wherein a full integration is performed for a few time steps at a coarser grid spacing.
3.  Level 3 testing is scientific validation of use cases at the target grid spacing.

Tests under Levels 1-2 have a faster turnaround time and are therefore part of continuous integration. Level 1 tests are performed at runtime comparing the outputs from the ported (GT4Py) and the reference (Fortran+) codes. Here, one sets an acceptable tolerance based on experience, typically $10^{-12}$ or smaller for double precision computations. The level 2 test is the probabilistic testing described in Probtest (2023), and used in Giorgetta et al. (2022) and Lapillonne et al. (2025). It works along the lines of early work by Rosinski and Williamson (1997) to check if the error of the ported code falls within the expected error growth of initial perturbations in the reference code.

Figure 5 shows an example of a Level 2 test performed on 10m diagnostics for Global aquaplanet use case. The relative error in the ported code is the difference between the outputs produced by the GPU binary of the ported code and the CPU binary of the reference code. Tolerance statistics (mean, max, and min) are estimated from a difference of the outputs from an unperturbed CPU simulation and 19 perturbed CPU simulations (Dipankar et al., 2025). To pass the test, relative errors in the ported code (dashed lines in Fig. 5) must be less than the acceptable tolerance indicated by the solid lines.

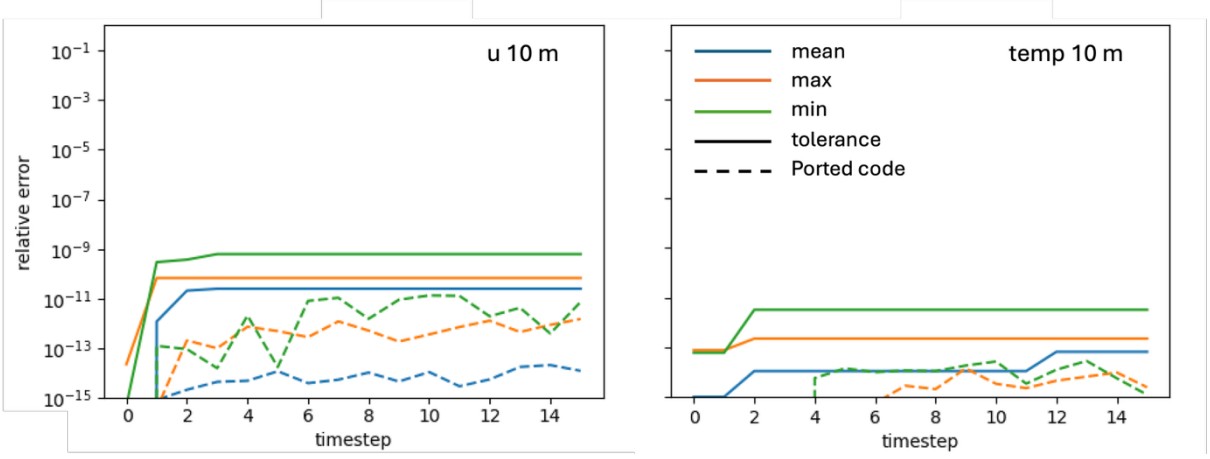

Figure 5 Probtest test applied on the aquaplanet use case for the indicated variables. the solid lines indicate the accepted error in the variable and the dashed lines show the error in the ported code. the colors indicate various measures of the error.

Level 3 testing is the final round in which the entire code base is subjected to a scientific use case
of varying complexity and the results are then validated against reference. The reference can be
a more mature model, for example in the case of idealised aquaplanet experiments, or
observations in realistic configurations. This testing is performed by the experts who have a better
understanding of the physical processes that the model simulates. In addition, level 3 testing also
help identify issues in the model code and the entire computing platform, which are not captured
in the level 1 and 2 tests. Validation of the new dynamical core is discussed further in Section 6.
Finally, it is important to note that in addition to the three-tiered testing of refactored ICON
mentioned above, GT4Py, as a library, undergoes its own testing (Paredes et al., 2023).

# 5    Computational Performance

Since performance is one of the key motivations to our development, the model with GT4Py
dynamical core is compared against the Fortran+ version of the model (Lapillonne et al., 2025) to
demonstrate feasibility of the approach. The two (user) codes are same except for the dynamical
core. All simulations are performed using full physics except for convection, gravity-wave drag,
and subgrid-scale orography drag parameterization as in prescribed SST simulations discussed in
section 6.3. Simulation length is 24 hours using a fixed time step of 22 seconds and 120 vertical
levels. Radiation is called every 15 mins, outputs are turned off, and only the integration time is
measured and are available at Dipankar et al. (2025). Simulations are performed on the Swiss
National Supercomputing Centre (CSCS) ALPS infrastructure on NVIDA GH200 processors.

## 5.1    Benchmarking

We consider the strong- and weak-scaling of the implementation, as well as the performance
comparison with the reference Fortran+ implementation. Figure 6 shows the weak scaling. There
is some degradation in the performance for two reasons: first, the halo region becomes
proportionally larger than the process-local domain as the latter shrinks with increasing number
of GPUs. Secondly, there are many more MPI processes communicating, which leads to more load
imbalance and, thus, synchronization overhead. Time reporting reveals that the latter is primarily
responsible for the increased overhead.

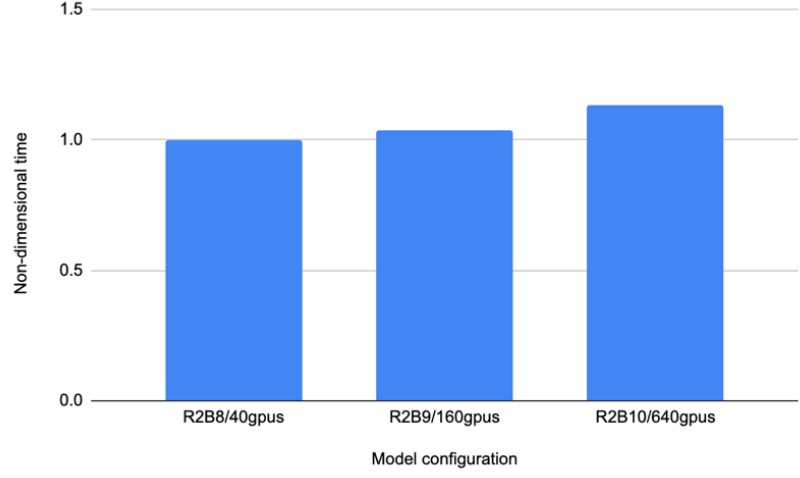


Figure 6: Demonstrating weak scaling for the grids R2B8, R2B9, and R2B10 which correspond to a horizontal
grid spacing of around 10 km, 5 km, and 2.5 km, respectively.

Figure 7 shows strong scaling comparison between the GT4Py (solid lines) and the Fortran+ (dashed lines) versions, not only for the dynamical core (yellow) but also for the full code during the time loop (blue). The GT4Py dynamical core performs about 10% faster than the Fortran+ near the GPU numbers where the memory required by the simulation configuration just fits. The differences between the two, however, reduces with increasing number of GPUs. Both implementations indicate an asymptotic limit to the strong scaling over the GPUs, which is a known issue (Giorgetta et al., 2022) and is understood to be due to the decreasing GPU occupancy. This implies that at even higher GPU numbers a CPU implementation, which has much better strong scaling, may outperform the GPU implementation. It is therefore crucial to choose a "just-fits" memory configuration (here 40 GPUs), which offers the maximal occupancy.

Interestingly, the timings difference in the dynamical core does not explain the larger performance difference in the overall time loop, even though the former is the only component which is different in the two implementations. A careful study of all the component timings suggests that MPI synchronization overhead is less in the new model, which also has an effect within the physical parameterisations.

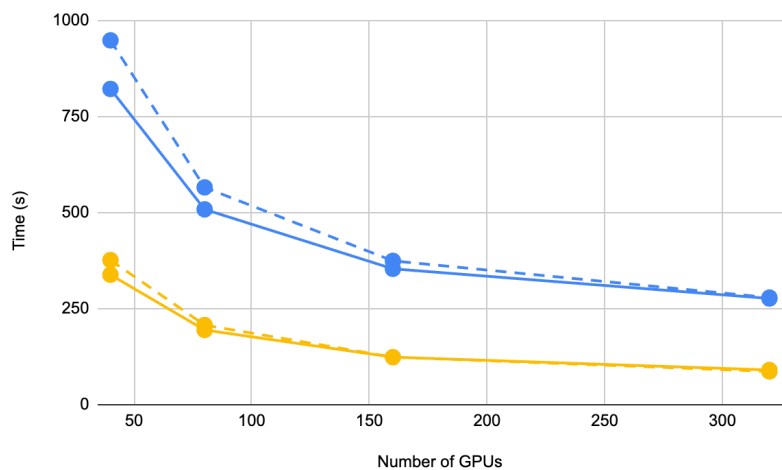

Figure 7: The overall timings (blue) of the R2B8 (10km) grid indicates an asymptotic limit to strong scaling for both the Fortran+ (dashed blue) and GT4Py (solid blue). The dynamical core (yellow) has a similar asymptotic limit. Near the "just-fits" configuration of (40 GPUs), the GT4Py (solid yellow) is about 10% faster than the Fortran+ (dashed yellow) implementation.

Figure 8 shows strong scaling in SDPD metric for R2B10 (2. 5 km) simulations. As indicated in the previous figure, there is an asymptotic limit due to the decreasing GPU occupancy, such that there is little reason to more than quadruple the GPU configuration to improve throughput. At 2560 GPUs, which is about a quarter of the CSCS ALPS infrastructure, the throughput we get is about 213 SDPD at R2B10 using 120 vertical levels and without coupling to the ocean. With an aim of 365 SDPD, one clearly sees the need of a radical change in the model design and compute architecture to further improve the computational performance, while constraining the energy consumption, as also discussed in Adamidis et al. (2025).

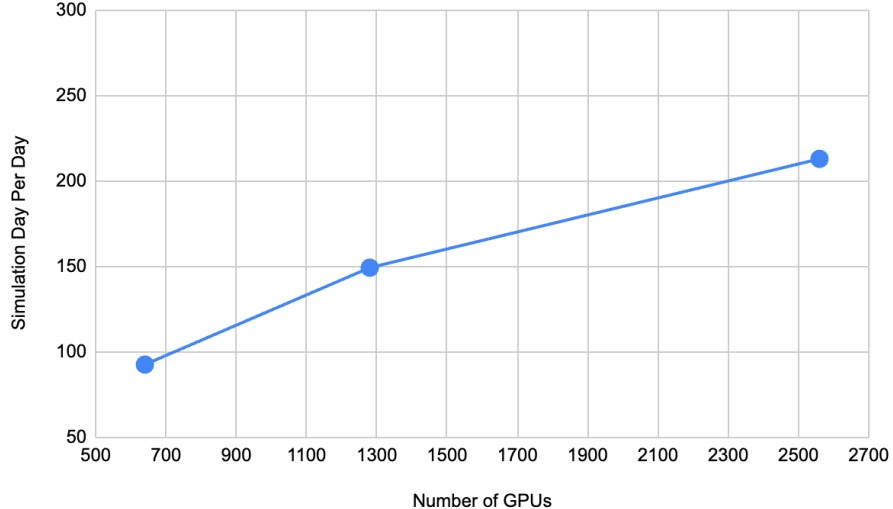

Figure 8: The simulation throughput in simulation days per day (SDPD) on R2B10 grid using 120 vertical levels with the GT4Py dynamical core. The strong scaling indicates some speedup but it quickly reaches an asymptote as the GPU occupancy decreases. Again, the just-fits configuration (here 640 GPUs) should be chosen to reap the maximal benefit from the GPU.

## 5.2   Future potential for further optimization

The performance of the GT4Py dynamical core outperforms the Fortran+ reference version by roughly 10%. There is some satisfaction in this result: the performance of the latter has culminated after years of optimization, while the former can still benefit from ongoing optimizations in the GTFN and DaCe backends. There is also the opportunity to fuse stencils, potentially increasing overall dynamical core performance. This work is currently ongoing and will be reported in future publications.

While it is difficult to estimate the potential limit for optimization in the backends, one guide could be the hand-written CUDA implementation of the dynamical core written by Nvidia developers [Pers comm., Alexeev D]. This so-called speed-of-light (SOL) implementation indicated that the overall dynamical core could potentially be sped up by a factor of two or more. The GTFN/DaCe backend development is leveraging the lessons from the SOL implementation.

## 6   Validation

Model validation is guided by a suite of scientific use cases of increasing complexity, including global aquaplanet (atmosphere-only), global uncoupled (atmosphere–land), and global coupled (atmosphere–land–ocean) simulations. Additionally, limited-area applications targeting numerical weather forecasting and regional climate modelling are considered. To date, the current model version has been employed in global aquaplanet and global uncoupled simulations. In this work, we present selected results from these simulations (see Dipankar et al., 2025 for data), with comprehensive analyses to be reported in separate publications.

Simulations are performed using the NWP (Numerical Weather Prediction) scientific configuration described in Zängl et al. (2015) and Prill et al. (2023) and the code version in Dipankar (2025) except for the use case in section 6.2 that uses XPP (eXtended Predictions and Projections) scientific configuration described in Müller et al. (2025) and made available in Müller et al. (2024). Both configurations use ecRAD (Hogan and Bozzo, 2018; Rieger et al., 2019) radiation scheme and single-moment bulk scheme of Seifert (2008). The turbulence schemes in NWP configuration is

based on Raschendorfer (2001) whereas XPP uses Mauritsen et al. (2007). The land surface
scheme in the NWP configuration is TERRA (Heise et al., 2006) whereas it is JSBACH (Reick et al.,
2021) in XPP. ICON dynamical core employs an ad hoc treatment of three-dimensional turbulence
at this scale by treating horizontal numerical diffusion using Smagorinsky (1969) closure.
For horizontal grid spacing of 5 km and smaller, deep convective parametrization, gravity-wave
drag, and subgrid-scale orography drag have been turned off except for the use case in section 6.2
where the subgrid-scale orography drag is kept on.
It should be noted that the atmospheric configurations used here are different from the ICON
Sapphire configuration described in Hohenegger et al. (2023). This is particularly true for the
treatment of sub-grid processes for land and atmosphere. The scientific details of dynamical core
is identical in all the configurations.

## 6.1   Global aquaplanet

State-of-the-art climate models struggle to accurately represent the intensification of
extratropical cyclones as well as the position, intensity, and tilt of the stormtracks, especially over
the North Atlantic. This is the case in terms of both tropopause-level Eddy-kinetic energy and
surface level cyclone frequency. Schemm (2023), by performing a nested 5 km simulation over the
storm track region, has shown that an increase in horizontal grid spacing towards storm-resolving
regime has the potential to lead to more tilted and poleward positioned stormtracks downstream
of a sea-surface temperature (SST) front. Validating this hypothesis led to the first use case for
EXCLAIM.
Following Schemm (2023), global aquaplanet is set up with 10 K SST anomalies in the shape of
ellipsoids in both hemispheres. The resulting SST mimics the Gulf Stream plus the land-sea
contrasts along the east coast of North America- replicating the formation of the North Atlantic
storm track downstream of the Gulf Stream. The simulations are performed on grids R2B10 (2.5
km), R2B07 (20 km), and R2B05 (80 km) using 90 vertical levels. The initial condition and SST for
each of these grids are perturbed to generate 3 ensemble members each of which one year-long
post spin up. Here, we only show the results from the finest (R2B10) and the coarsest (R2B05) grid
spacing simulations to demonstrate the effect of storm-resolving global simulations.

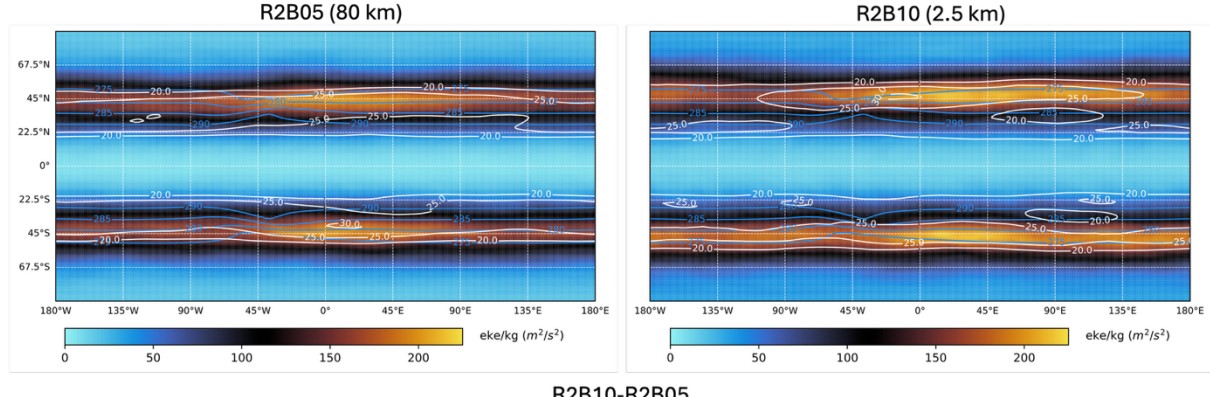

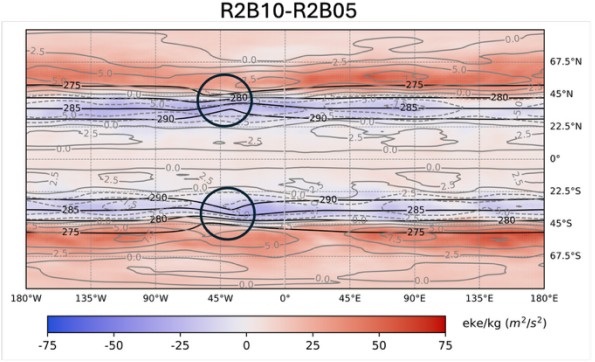

Figure 9: Mean state of the jet in aquaplanet simulations for indicated grids (top) and their difference (bottom). EKE is shaded and the horizontal wind speed is depicted in white (grey) contours (dashed negative) starting at 20 m/s with a spacing of 5 m/s (2.5 m/s) in top (bottom) panel. The blue (black) contours are SST in top (bottom) panels starting at 275 K with a spacing of 5 K. Region around the SST fronts are circled in the bottom panel.

Figure 9 shows the ensemble mean of eddy kinetic energy (EKE) and horizontal wind speed averaged over the simulation period and vertically between 450 hPa and 250 hPa for the two grids and their differences. Indeed, the resolution jump affects both the mean state of the jet and its variability. The annual mean wind speed fields show that the jet is stronger and more poleward in the R2B10 than R2B05 simulations. The lower panel shows that this difference is particularly marked downstream of the SST front. This is also the region where the difference between the storm tracks marked by the difference in EKE is strongest. As for the mean wind speed, EKE is larger and more poleward for the R2B10 ensemble mean. This is especially true in the 180° downstream of the SST front, where the mean jet is also most poleward. The correlation between the differences in the mean state of the jet and the storm track point toward the importance of eddies for shaping the mean jet: The jet is strongest where EKE is, and the differences between the mean states of the R2B10 and R2B05 jets are strongest where the differences in EKE are.

Overall, these findings are in line with Schemm (2023) showing the potential of km-scale simulations to better represent stormtracks compared to a typical climate model. Further study will focus on the impact of grid spacing on the most extreme winds in the jet stream to enhance our understanding of the mean jet stream representation and its variability.

## 6.2  Global uncoupled with idealized SST perturbations

Equilibrium climate sensitivity (Charney, 1979) describes the global mean temperature increase following a doubling of $CO_2$ concentrations. It is a key parameter for assessing the planet's vulnerability to climate change. However, its uncertainty range has remained approximately constant over the past 40 years. Cloud feedback is one of the main sources of uncertainty (Sherwood et al., 2020). Current state-of-the-art Coupled Model Intercomparison Project (CMIP)

type climate models heavily rely on parameterisations, which are known to be a major contributor
to the uncertainty. As model grid spacing increases, more essential processes can be directly
resolved and represented by explicit physical equations. We evaluate the differences in feedback
processes in response to SST perturbations between coarse-resolution simulations with the full
set of parameterisations and high-resolution simulations without deep convective and gravity
wave parameterisations. We focus on idealised SST perturbations such as a warming patch in the
Western Pacific (GFMIP protocol, Bloch-Johnson et al., 2024) and realistic perturbation such as El
Niño.
Simulations are performed using the ICON XPP configuration targeted for seasonal and climate
simulations (Früh et al., 2022; Niemeier et al., 2023; Müller et al., 2025). A characterisation of the
km-scale setup can be found in Kroll et al. (2025). Two horizontal grid spacings are tested: R2B06
(40 km) and R2B09 (5 km), both with 150 vertical levels and a model top at 75 km. For the R2B06
configuration, all parameterisations are active whereas the parameterisations for deep
convection and gravity waves are switched off for R2B09.
Capturing the atmospheric teleconnections between the tropics and extratropics is especially
important for an accurate representation of feedbacks studied in this use case. For this, the
absence of the double Intertropical Convergence zone (ITCZ) bias is essential. The mean
precipitation bias of the 40 km and 5 km configuration against Global Precipitation Measurement
Integrated Multi-satellitE Retrievals for GPM (GPM IMERG; Huffman et al., 2019) shows that the
single strand ITCZ is expressed correctly (see Fig. 10). The skill of both configurations in capturing
the large-scale precipitation fields is comparable, however the regions of dominating biases shift.
For example, the 5 km setup exhibits an improved representation of precipitation over islands in
the Tropical Warm Pool, whereas the 40 km setup has reduced biases at the coast of India and
Burma. A detailed description is available in Kroll et al. (2025) and will be accompanied by an
analysis of the atmospheric feedback in a separate work.

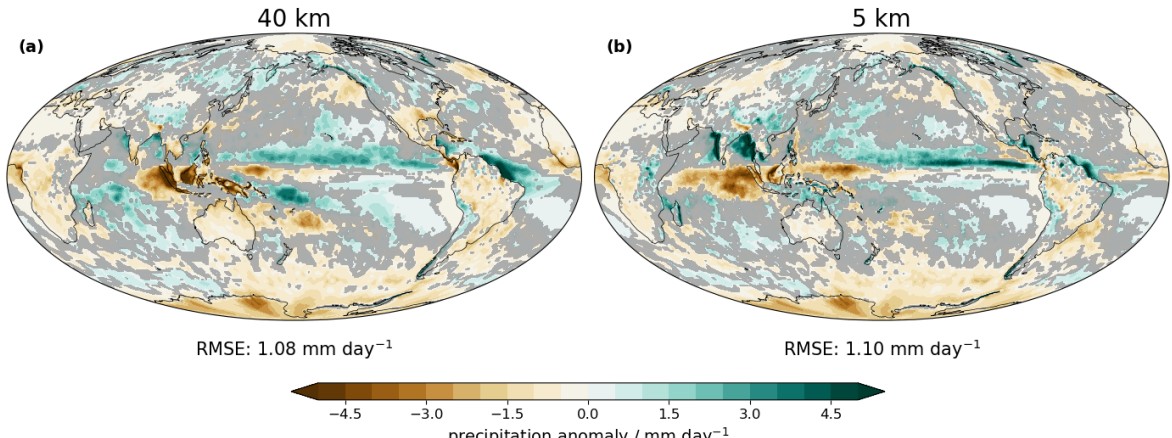


Figure 10: Two-year mean global precipitation bias with respect to the 2004-2010 average of IMERG
precipitation field for the (a) 40 km (R2B06) and (b) 5 km (R2B09) configurations. Statistically significant
differences, based on a two-sided z-test at α = 0.1, are shown; insignificant regions are grayed out. All data
was remapped to a grid using 140 km spacing for better comparability. The Global Root Mean Square Error
(RMSE) for both configuration is shown below the maps.

## 6.3  Global uncoupled with realistic SST

Global uncoupled simulations with realistic prescribed SSTs served as the first realistic use case. These configurations are critical to the success of envisioned digital twins of Earth system (Bauer et al., 2021; Hazeleger et al., 2024) to aid adaptation decisions for the changing climate at a community level, and for a better understanding of the Earth system in general. First studies have demonstrated the potential of global storm-resolving simulations in better representing key atmospheric processes that affect clouds and precipitation (Kuma et al 2024, Lee and Hohenegger 2024, Spät et al 2024) and their effectiveness in realising extreme precipitation (Wille et al 2024). The present use case complements these studies by analysing the vast amount of information contained in these simulations from a different perspective. Detailed analyses of the results will be reported in separate publications. Here we present some of the results, highlighting the potential of the current configuration.

The simulation is performed on R2B10 grid (2.5 km) using 120 vertical levels. The science configuration follows the protocol with a motivation to contribute towards DYAMOND phase – III globally coordinated experiments as described in Takasuka et al. (2024). The simulation is initialised by European Centre for Medium-Range Weather Forecasts (ECMWF) analysis data on 2020-01-20, 00UTC using European Space Agency Climate Change Initiative (ESA-CCI) SST at a horizontal spacing of 1/20° updated daily. Soil moisture is spun up through another 10-year simulation at 10 km horizontal spacing and then regridded to R2B10. The simulation is conducted for a period of 4 years starting from January 2020 to March 2024. The first two months are discarded for spin up.

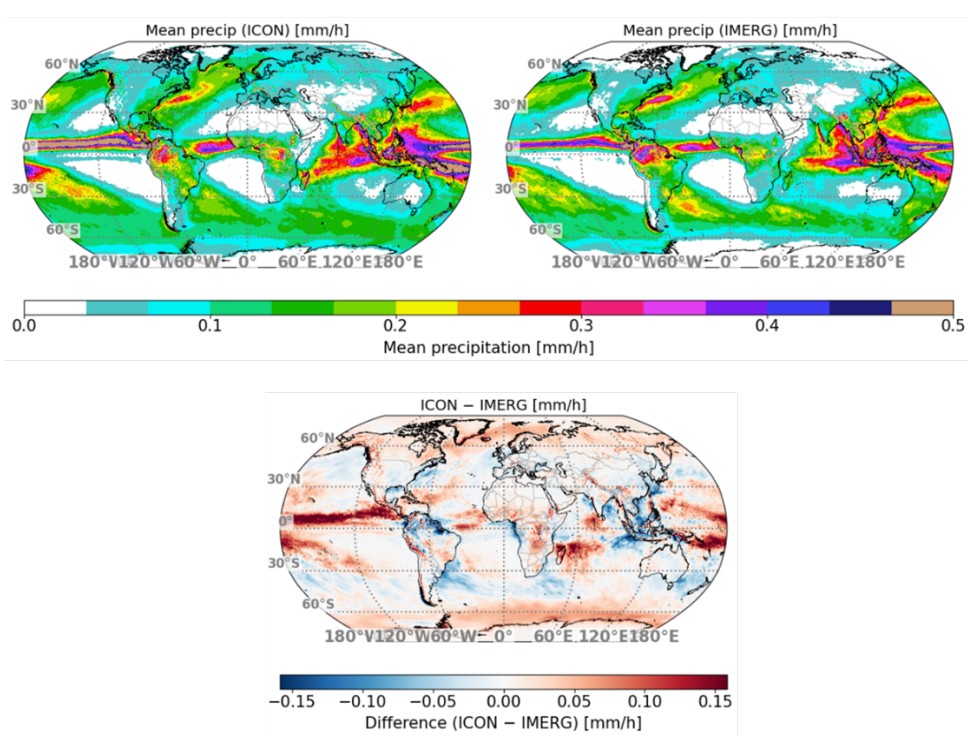

Figure 11: Mean precipitation in ICON (top left) and observation (top right). The difference is shown in bottom panel

Figure 11 shows the mean precipitation over the simulation period in ICON and GPM IMERG.
Broadly speaking, the precipitation pattern and amplitude are nicely captured. The equatorial
region, particularly over the ocean (as seen in the bottom panel), shows wet biases that warrant
improvement. Interestingly, the present configuration eliminates the underestimation of
equatorial rainband in the Indo-Pacific region, famously known as the double ITCZ (Inter Tropical
Convergence Zone) observed in ICON simulations using Sapphire configuration (see Fig. 5 in
Segura et al., 2025). The wet bias in the equatorial region is also apparent in the monsoon
rainbands discussed next.

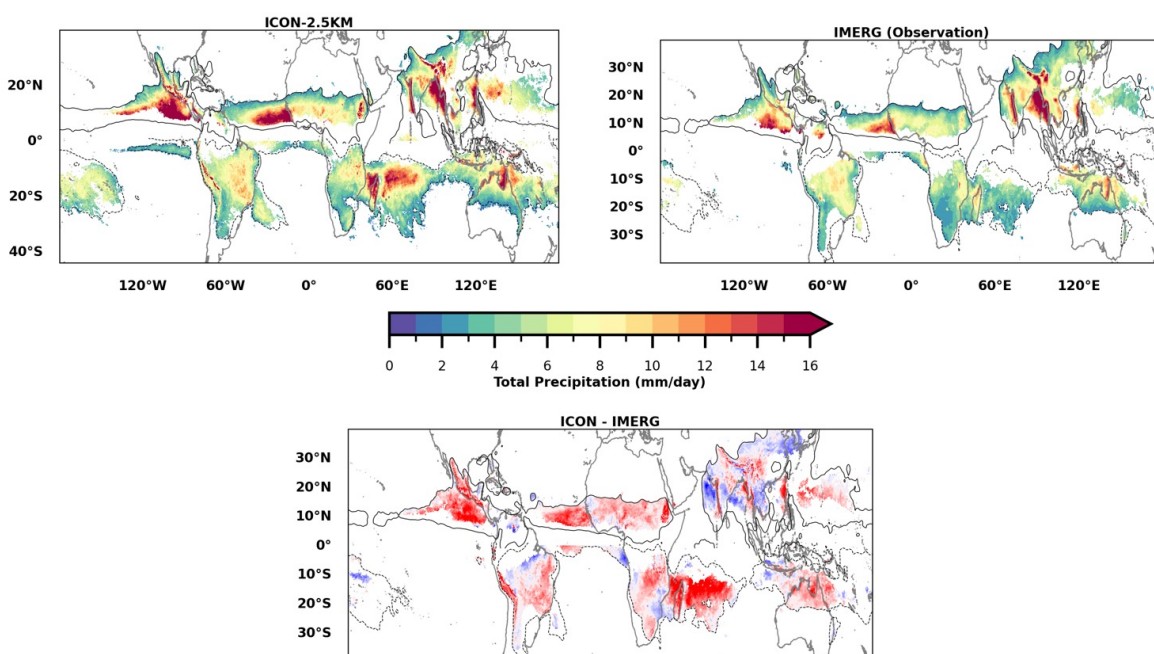


**Figure 12.** Global monsoon domains (shaded) are defined as regions where the difference between local
summer and winter precipitation exceeds 2 mm day⁻¹, and local summer precipitation contributes at least
55% to the total annual precipitation. The black contours indicate the 2 mm day⁻¹ threshold of summer-minus-
winter precipitation from (a) the ICON 2.5 km simulation, (b) IMERG observations, and (c) their difference
(ICON – IMERG).
The global monsoon system is a dominant feature in the tropical region exhibiting seasonal reversal
of winds while contributing significant amount of precipitation. The systems exhibit rich variety of
scales and their interactions ranging from large scale teleconnections to intraseasonal oscillations
and diurnal cycle. Figure 12 highlights the global monsoon domains, following the definition of
Wang & Ding (2008): regions where summer minus winter precipitation exceeds 2 mm day⁻¹ and
summer precipitation contributes at least 55% of the annual total.
The ICON 2.5 km simulation captures the major domains, including the South and Southeast Asian,
East Asian, West African, and North American monsoons, in broad agreement with IMERG
observations. Other regions with strong seasonality, such as South Africa, Central America, and
equatorial South America, remain debated in terms of their classification as monsoon domains
(Climate Change, 2021) and are not discussed further. The contours in Fig. 12 indicate the 2 mm

day$^{-1}$ summer–winter precipitation threshold, showing that ICON resembles IMERG but tends to overestimate precipitation over the Pacific Ocean and the American and Australian maritime regions. These areas, however, fail the 55% annual contribution criterion and thus are excluded as monsoon domains. The difference panel highlights a general wet bias in ICON relative to IMERG, with the exception of localized dry biases over South/Southeast Asia and East Asia. A more detailed analysis of the underlying dynamical and thermodynamical causes will be presented in other detailed study

# 7  Conclusion

This work presents the refactored atmospheric dynamical core of ICON written in GT4Py integrated within the existing Fortran-based infrastructure. The new implementation demonstrates performance slightly superior to the Fortran+OpenACC version that has been performance-tuned over time, while offering a cleaner and architecture-agnostic code base. The use of a Python-based DSL enables separation of concerns, facilitating portability across heterogeneous CPU-GPU computing platforms and laying the foundation for future model evolution.

The refactored code has been subjected to a comprehensive testing strategy, including unit-level verification, integration tests, and scientific validation. Preliminary results from global aquaplanet and uncoupled simulations demonstrate the model's ability to realistically capture key atmospheric processes, such as storm track dynamics and precipitation patterns, highlighting the potential of high-resolution global simulations to address persistent challenges in climate modelling.

While the current implementation already shows a modest performance gain over the reference, further optimizations, such as performance tuning using DaCe are expected to yield improvements but not enough to reach the goal of one simulation year per computational day. Poor strong scaling on CPU-GPU architectures appear to be the limiting factor. We believe that transition to a Python-based model infrastructure and components have the potential to allow for more radical changes towards achieving the performance goal in addition to improving user experience.

# 8  Acknowledgment

This work was supported by the EXCLAIM project funded by ETH Zürich and by the contributions from ESiWACE2 and ESCAPE-2. The authors thank the EXCLAIM core team members Christina Schnadt Poberaj and Tamara Bandikova for their inputs and suggestions.

Computing and data storage resources were provided by the Swiss National Supercomputing Center (CSCS) in Lugano via the projects: cwd01 (AD), cwp02 (MBK), cwp03 (PPK), cwp04 (CAK). CAK gratefully acknowledges support by the ETH Postdoctoral Fellowship Program.

# 9  Code and data availability

The ICON code used in the manuscript is available under a permissive BSD-3C license. Details on code availability and usage can be found at https://www.icon-model.org/.  The source code and the run scripts used for the global aquaplanet simulations and global uncoupled simulations with realistic SST are available at https://doi.org/10.5281/zenodo.17250248 (Dipankar, 2025). The source code and other relevant scripts for the global uncoupled simulation with idealized SST perturbations are available at https://doi.org/10.17617/3.UUIIZ8 (Müller et al., 2024).

Simulation and observation data used to generate the figures are available at https://doi.org/10.5281/zenodo.17317423 (Dipankar et al., 2025).

## 10 Author contributions

MB, AD, XL, and TCS outlined the development roadmap. AD led the manuscript. The rest of the authors have contributed equally to the manuscript and to the development.

## 11 Competing interests

The authors have no competing interests.

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
