# Peer review of "Toward Exascale Climate Modelling: A Python DSL Approach to ICON's (Icosahedral Non-hydrostatic) Dynamical Core (icon-exclaim v0.2.0)"

_EGUsphere, 2025_

## Author Comment (AC1)

**Reviwer 1**

This manuscript presents stage one of a multi-tiered plan to support heterogeneous (mixed CPU/GPU) architectures for running the ICON model. The authors utilize GT4Py, a domain-specific language, to modernize the ICON dynamics core from the existing Fortran code base. The outcome is a more performant code, which is also easier to read and develop compared to the equivalent Fortran implementation. The paper is well written and well reasoned, demonstrating promising results that are on par with the current state of GPU-ready Earth System modeling. I recommend that this manuscript be published, as I have only a few minor questions and technical corrections to suggest.

First, I want to commend the authors for their attention to (a) the hardware-based challenges that arise when running these models at scale, and (b) the importance of robust testing. In my experience, these topics are not typically the most exciting to discuss, but they are essential considerations       for       any       group       undertaking       a       similar       effort.

We sincerely thank the reviewer for taking time to reviewer the paper and for appreciating our work.

Minor Comments:

Introduction

1. Paragraph 3: It may be helpful to include node counts when discussing how much of the machine each example used. This additional detail would provide useful context, especially as future machines come online.

   We agree with the reviewer that additional details would be useful. However, adding them to the same paragraph would make the introduction quite involed with details on hardwares. We have therefore repharsed the paragraph with only relevant details and have suggested interested readers to look at Table 1 in Klocke et al. (2025) for more details. Please see lines 62-70 in the revised version.

   Paragraph 6: As noted above, I appreciate the discussion highlighting barriers to running these models at scale.

   Thanks again!

Section 2

1. Not strictly necessary, but it may add valuable context for readers if the authors note that Fortran compiler support is increasingly being deprioritized by vendors, which makes supporting legacy codes on new machines more challenging.

   We appreciate the reviewer for reminding this point. It has been added in the revised introduction on line 89 when discussing softare reliability.

Section 3

1. I may have missed it, but it was unclear whether the plan is to transition entirely away from Fortran after deliverable 3. Could the authors clarify how much of the original Fortran code is expected to remain in the model (e.g., 10%, 25%, or more)?

Thanks for asking this question. We do aim for a Fortran-free driver/infrastructure code in deliverable 3. The no-Fortran infrastructure so far is complete for idealized simulations using dynamical core alone.

As for the model components, we will (likely) keep land-surface parameterization and Ocean in Fortran in the foreseable future.

Section 4

1. General comment: The authors should verify that each "Listing" is correct and that the code blocks would work as expected.

   Thanks for pointing it out. The listings have been checked again. They are correct in what they represent but we do not expect them (e.g., Listing 1) to work by simply copy and pasting.

2. Section 4.3: If I understood correctly, the ported code was tested to within a tolerance error, and bit-for-bit (BFB) agreement was not strictly enforced. Was any BFB enforcement attempted during porting? If not, could the authors justify their decision not to enforce BFB?

   Thanks for asking this question. Enforcing BFB agreement was not deemed as a sustained testing strategy within the project since we had intentions to combine stencils into larger GT4Py operators/programs for performance tuning through DaCe. Maintaing such a debugging mode would have been difficult. That said, we did attempt BFB agreement during dynamical core porting- we used the -iEEE flag to prevent Fortran from doing non-IEEE 754 compliant transformations of floating point computations. We also switched off fused multiply adds (FMA) on both sides (Fortran and generated CUDA code in gt4py). BFB was achieved in the vast majority of stencils except for a few, even though the codes were correct.

3. Figure 5: Did the authors conduct experiments with runs well beyond 15 timesteps to confirm that the relative error does indeed stabilize?

   Yes, we did and the errors did stabilize.

4. Figure 7: Did the authors examine this data using a log-log plot? If so, was the observed trend not quite linear?

   You are right. Here's the log-log plot for your reference.

---

## Author Comment (AC2)

Reviewer 2

This is a clear well written paper describing a gt4py implementation of the ICON dynamical core, running in the existing ICON Fortran modeling system, enabling k-scale atmospheric simulations on the ALPS GPU supercomputer. The authors describe their porting approach, including thorough testing from the kernel level up to full physics simulations. They provide a sober analysis of the potential of GPUs and their strong scaling limitations.

We sincerely thank the reviewer for taking time to reviewer the paper and for appreciating our work.

I only have minor comments:

1. Section 4.3: what is "the implementation of horizontal blocking"? Does that refer to the loop blocking in the Fortran loops, (which was removed in the Python code?)

Yes. We have also adjusted the sentence (see line 328) to include "horizontal loop blocking" so that it is clearer.

2. Section 4.3: "...testing is tricky as the results are different due to rounding..."

The authors have a good port testing strategy in the presence of roundoff error, but this statement implies that these rounding differences are unavoidable. The E3SM dycore porting work (Bertagna et al. GMD 2019 and Bertagna et al. SC2020) showed that it is possible to obtain BFB agreement between CPUs and GPUs with careful coding, allowing for a different porting approach which simplifies some aspects of code porting.

We agree with the reviewer that enforcing BFB agreement for debugging purposes is helpful. We used it during dynamical core porting by using the -iEEE flag to prevent Fortran from doing non-IEEE 754 compliant transformations of floating-point computations. We also switched off fused multiply adds (FMA) on both sides (Fortran and generated CUDA code in gt4py). BFB was achieved in the vast majority of stencils except for a few, even though the codes were correct. However, since the compute stencils were continuously being combined into larger GT4Py operators/programs for performance tuning through DaCe, we did not think that maintaining BFB reproducibility would add any value in longer run.

3: Section 5.1:

For the final model, I assume all significant code is running on the GPUs, with the dycore using gt4py and the physics using openACC. I believe this is implied, but I didn't see it clearly stated. Were there any software challenges running the two different GPU programming models in the same executable?

Thanks for pointing it out. We have adjusted the paragraph (see lines 371-374) to make it clear. As for running the OpenACC and GT4Py codes in single binary- this was not much of a problem since all memory allocations took place in the OpenACC side. The largest change was to make the OpenACC code work with no-horizontal-blocking.

4. Line 400: "GT4Py synchronization"

I know of two types of synchronization: across MPI nodes, as well as synchronization among thread teams running on the GPU. Which is this referring to?

Thanks, again. It was a mistake. It was the MPI synchronization that took lesser time in the new model. The revised mansucript has been updated accordingly. See line 405.

5. Section 5.1

How does the gt4py code compare with the Fortran code on CPUs? It would be interesting to add CPU-only performance numbers to Figure 7.

Thanks for showing interest on CPU benchmarks. Unfortunately, the structure of the current mansucript is such that the authors did not consider adding CPU benchmarks here. However, we intend to write a mansucript on deliverable 2 very soon, where the focus will be on (performance) portability. We believe that the CPU benchmarks better belong there.

---

## Author Comment (AC3)

**Reviewer 3**

The paper is about a re-implementation of the ICON dynamical core using a domain-specific language embedded into Python called GT4Py. The work is carried out in the EXCLAIM project for which the paper presents the outcomes of the first phase. Described is the porting approach when rewriting the dynamical core into GT4Py, the testing strategy during the work, an evaluation of the computational performance and the scientific validation of the new code.

I found the paper was written in an accessible way, with a clear and sensible structure that covers all relevant angles of this development. The achieved milestone of the dynamical core rewritten in GT4Py is a remarkable achievement and the approach that utilised a very thorough testing procedure was well designed to avoid mistakes as much as possible. I would recommend a few minor edits to improve the overall presentation, which I list below with reference to the relevant sections of the text:

We thank the reviewer for taking time to reviewe the paper and for appreciating our work.

The abstract presents a specific throughput number but without specifying for what configuration or resolution. I would either add more details or leave it at the statement that the GT4pPy core exceeds ICON OpenACC performance without giving a specific number.

Thanks for point it out. The abstract has been accordingly adjusted.

The overview of current performance numbers in the paragraph in ll. 58ff is a wild mixture of very different configurations and resolutions. The intention is likely to take stock of how close current ESMs get to the 1 SYPD target, but this gets lost in the presentation. I would suggest to make this a little more focussed, ideally using a more like-for-like comparison. Moreover, most numbers are presented without references (NICAM, IFS-FESOM, ICON@1.25km). Some should stem from the GB submissions (https://dl.acm.org/doi/10.1145/3712285.3771789 and https://dl.acm.org/doi/10.1145/3712285.3771790) but it is irritating to see them published in this preprint before the availability of the original papers, particularly when no reference is given.

Thanks, again. The paragraph has been adjusted. Please see lines 62-70 of the revised manuscript.

In l. 97ff the three-phase nature of EXCLAIM is mentioned but no further information about the planned content of phases 2 and 3 is provided. Does this correspond to the deliverables shown in Figure? In the same paragraph, it is stated that the rewrite is "driven by the existing Fortran driver", which I did not understand until much later. Maybe this could be described in a form that makes it clearer that it is embedded into the existing Fortran framework, replacing calls to the dynamical core routines.

Done. Please see lines 97-100 in the revised manuscript.

Figure 1 is a useful illustration of the GT4Py code generation pipeline. I suspect not every reader may be familiar with therein used acronyms "GTIR" and "GTFN", which could be spilled out in the caption. GTIR is clarified later in the text but GTFN remains unclear.

Thanks for poiting it out. Figure 1 caption has been modified to explain the acronyms.

In l. 154f, three execution modes for running GT4Py are mentioned. Which of these are used here? Given that this is embedded into Fortran, I suspect this requires AOT?

Yes. A sentence has been added in the revised text to make it clear.

I did not immediately recognize the term "Fortran+" in l. 169 as the introduction of nomenclature. Maybe putting this in quotes would be helpful?

Done.

The description of the refactoring work in Sec. 4 is well written with an appropriate level of detail. The formatting of Listing 2 is unfortunate, with a page break between the listing and the caption - this should be rectified before final publication.

I agree on the readability angle in l. 278 but I did not understand the reason why only Python should allow in-line documentation through docstrings. I would argue that this could be done in any language, including Fortran.

You are right. We have modified the sentence to indicate that it *easier* to do so in Python.

The resolution of Fig. 4 seems a little low, it shows some artifacts in my print-out.

Thanks, again. Figure has been updated.

The hierarchy of testing levels appears well thought-out and seems effective to cover testing from a fine-grained stencil-loop level to full system regression. How much of this is automatic and when is it run? How expensive are these tests (in core-h or similar)?

Testings in level 1 and 2 are automatic and are part of the CI. For smaller experiments, it takes about 90 seconds for level 1 and about 30 seconds for level 2 for each experiment. Level 3 is full scientific simulation, which takes long and is only performed during this development phase. We do not expect to maintain it once the development is over.

The presented performance numbers are promising. However, in Fig. 7 either the plot colours or caption are wrong. The caption claims "GT4Py (dashed yellow) is about 10% faster than the Fortran+ (solid yellow)", while the plot suggests this to be the other way round. For the blue colours, dashed/solid seem to be reversed, so I suspect this may simply be a mistake in the caption.

There was indeed a mistake in the caption. Thanks for spotting it.

Given the substantial performance speed-up claims from the speed-of-light implementation: is there a specific pattern/generic improvement that accounts for this improvement? Or is it a mixture of several different changes?

In SOL implementation they used several tricks in addition to writing the code in CUDA C++ and inlining/fusion. The overall gain seem to be a result of all these combinations.

Since I'm not an expert on the scientific evaluation presented in Sec. 6, I cannot give a substantial feedback to this part.